# The Role of Xenobiotics and Anelloviruses in Colorectal Cancer: Mechanisms and Perspectives

**DOI:** 10.3390/ijms26094354

**Published:** 2025-05-03

**Authors:** Francisco Aguayo, Julio C. Tapia, Gloria M. Calaf, Juan P. Muñoz, Julio C. Osorio, Matías Guzmán-Venegas, Carolina Moreno-León, Jorge Levican, Cristian Andrade-Madrigal

**Affiliations:** 1Laboratorio de Oncovirología, Departamento de Ciencias Biomédicas, Facultad de Medicina, Universidad de Tarapacá, Arica 1000000, Chile; cejulio704@gmail.com (J.C.O.); maguzmanvenegas@gmail.com (M.G.-V.); carolinajohanamoreno@gmail.com (C.M.-L.); cristian.andradem@gmail.com (C.A.-M.); 2Laboratorio de Transformación Celular, Instituto de Ciencias Biomédicas, Facultad de Medicina, Universidad de Chile, Av. Independencia 1027, Santiago 8380453, Chile; 3Instituto de Alta Investigación, Universidad de Tarapacá, Arica 1000000, Chile; gmcalaf@academicos.uta.cl; 4Laboratorio de Bioquímica, Departamento de Química, Facultad de Ciencias, Universidad de Tarapacá, Arica 1000007, Chile; jpmunozb@academicos.uta.cl; 5Department of Microbiology, Icahn School of Medicine at Mount Sinai, New York, NY 10029, USA; jelevican@gmail.com

**Keywords:** colorectal, cancer, anellome, environmental, anellovirus, colon

## Abstract

Xenobiotics are non-natural chemical compounds to which the human population is exposed. Chronic exposure to certain xenobiotics is associated with various diseases, including cancer development. Anelloviruses (AVs), including Torque Teno Virus (TTV), Torque Teno Mini Virus (TTMV), and Torque Teno Midi Virus (TTMDV), are ubiquitous viruses found in the general population. As no disease has been definitively associated with AVs, they are sometimes referred to as “viruses awaiting a disease”. This review explores the potential roles of xenobiotics and AVs in colorectal cancer (CRC) development and suggests a potential interplay between them. Evidence suggests an association between certain xenobiotics (like pesticides, cigarette smoke components, and dietary factors) and CRC, while such an association is less clear for AVs. The high prevalence of AVs suggests these infections alone may be insufficient to disrupt homeostasis; thus, additional factors might be required to promote disease, potentially including cancer.

## 1. Introduction

Colorectal cancer (CRC) presents a significant global health challenge due to its high incidence and mortality rates [1]. While often considered a disease of developed countries, developing nations are also seeing increased CRC-related deaths [2,3]. CRC progression is often slow, with 25% of patients having metastases at diagnosis, and nearly 50% of those undergoing tumor resection developing metastases within five years [4].

Human populations are exposed to xenobiotics—chemical substances foreign to the body, such as environmental pollutants and industrial chemicals [5]. Compounds like polycyclic aromatic hydrocarbons (PAHs) and pesticides are common environmental contaminants [6,7]. Xenobiotics are implicated in CRC development through mechanisms potentially including DNA damage, inflammation, and microbiota changes, possibly influenced by dose, genetics, and lifestyle [8,9,10,11,12].

Alongside environmental factors, the role of the human virome, particularly anelloviruses (AVs), is gaining attention. AVs are highly prevalent but lack a confirmed pathogenic role, suggesting additional factors may be needed for them to contribute to diseases like cancer. This review examines the evidence for the roles of both xenobiotics and AVs in CRC and explores the perspective that they might interact in its development [13,14,15].

## 2. Background: Altered Signaling Pathways in Colorectal Cancer

Genetic evidence regarding CRC has demonstrated that, in most cases, the initial oncogenic mutation affects the Wnt/β-catenin signaling pathway. This pathway plays a critical role in development, stem cell maintenance, and tissue regeneration [16,17], but its dysregulation is linked to the development and progression of multiple cancers across various species [18]. Canonical Wnt/β-catenin pathway activity is regulated by targeting β-catenin into a multi-protein complex formed by the tumor suppressors Axin, APC (Adenomatous Polyposis Coli) and GSK3β, the latter ultimately phosphorylating β-catenin and targeting it for ubiquitination and subsequent proteasomal degradation [19]. Most CRC tumors contain either deletions in APC or mutations in β-catenin [20]. Since APC regulates β-catenin degradation, the inability of APC to promote its degradation is the main suggested cause of nuclear β-catenin accumulation and aberrant activity in CRC cells [21]. β-Catenin associates with transcription factors of the Tcf/Lef family, thus controlling the expression of genes that contribute to proliferation, resistance to apoptosis, and ultimately metastasis [20,21]. However, it was recently demonstrated that transcriptional activity promoted by β-catenin and TCF/LEF family members is not required for the survival and propagation of some human colorectal cancer cell lines [22]. On the other hand, canonical activation of the Wnt/β-catenin pathway leads to the inhibition of GSK3β, the lack of phosphorylation of β-catenin, its stabilization, and accumulation in the cytosol, followed by its increased nuclear localization [23,24].

Besides GSK3β, several other protein kinases, such as Akt, have been implicated in the upregulation of the Wnt/β-catenin pathway. Akt participates in several processes relevant to cancer progression, including proliferation, survival, and angiogenesis [25]. Akt upregulates this pathway by phosphorylating and inhibiting GSK3β, leading to several traits that promote cancer progression, such as decreased E-cadherin expression (a typical EMT marker), cancer cell detachment, and metastasis [26]. E-cadherin, a crucial cell adhesion protein, plays a vital role in maintaining cell–cell junctions; its downregulation contributes to the breakdown of these junctions, facilitating invasion and metastasis. Additionally, Akt can also promote the transcriptional activity of β-catenin through its direct phosphorylation of β-catenin [27].

Protein kinase CK2 is another enzyme involved in the upregulation of the Wnt/β-catenin pathway [28,29]. CK2 is abnormally elevated in a wide variety of tumors, which is associated with growth and proliferation [30,31]. CK2 was identified as a metastasis-associated gene in a proteomic study of CRC cell lines exhibiting varying metastatic potential [32]. Nuclear CK2 overexpression has been correlated with a poor prognosis in CRC patients [33]. CK2 also promotes β-catenin stability by direct interaction and phosphorylation, which protects it from being targeted by the negative-regulatory complex formed by Axin/APC/GSK3b, thereby enhancing its stability and transcriptional activity [34,35,36]. CK2 also indirectly upregulates the Wnt/β-catenin pathway by phosphorylating Akt. This, in turn, promotes β-catenin activity and invasiveness in a phosphorylation-dependent manner [37,38,39]. While this section focuses on key pathways like Wnt/β-catenin, Akt, and CK2 known to be altered in CRC, it is important to note that other signaling cascades, including TGF-β, MAPK/ERK, and cholinergic signaling pathways, also play significant roles and will be discussed in subsequent sections where their relevance to specific xenobiotic actions or potential viral interactions becomes pertinent.

## 3. Environmental Factors and Colorectal Cancer

### 3.1. Xenobiotics: Tobacco Smoke

Tobacco smoke, a complex mixture containing thousands of chemicals [40], is strongly associated with colorectal cancer (CRC), although the precise mechanisms are multifaceted [41]. Key carcinogenic processes involve DNA damage through adduct formation, the dysregulation of critical cellular signaling pathways, and the promotion of inflammation [41]. A primary mechanism linking tobacco smoke to CRC is the formation of DNA adducts by various carcinogens, leading to genetic instability. Prominent examples of these carcinogens include N-nitrosamines like 4-(methylnitrosamino)-1-(3-pyridyl)-1-butanone (NNK) and N′-nitrosonornicotine (NNN), polycyclic aromatic hydrocarbons (PAHs) such as benzo[a]pyrene (BaP), and aromatic amines like 2-amino-9H-pyrido [2,3-b]indole (AαC) [42,43,44]. Following metabolic activation, these compounds or their metabolites covalently bind to DNA [42,43,44]. These adducts, if unrepaired, can cause mutations during DNA replication, potentially affecting critical genes involved in CRC, such as oncogenes and tumor suppressor genes [44]. The presence of such adducts serves as a biomarker of exposure and increased CRC risk [45]. For instance, NNK is a potent inducer of tumors via adduct formation [42,43], while BaP metabolites also form stable adducts and can induce oxidative stress, further exacerbating DNA damage [44]. Aromatic amines like AαC have been shown to induce aberrant crypt foci, precursor lesions for CRC, in animal models [46,47]. While the specific adducts may differ, the common outcome is damage to the genome, a crucial step in carcinogenesis [43].

Furthermore, tobacco smoke components disrupt multiple signaling pathways essential for intestinal homeostasis, thereby promoting CRC development and progression. Current evidence indicates impacts on pathways including, but not limited to, PI3K/Akt and TGF-β signaling [48]. The PI3K/Akt pathway is crucial for cell survival, proliferation, and metabolism, and tobacco carcinogens can contribute to its aberrant activation [49]. Dysregulation of the PI3K/Akt/mTOR network is frequent in human cancers, including CRC, where the overexpression of Akt and loss of the tumor suppressor PTEN are common findings [50,51]. The activation of this pathway by tobacco components can inhibit apoptosis and promote tumor cell survival and proliferation. The TGF-β pathway also plays a complex role, often acting as a tumor suppressor in early cancer stages but potentially promoting invasion and metastasis later. Tobacco compounds can disrupt TGF-β signaling, contributing to tumorigenesis [48]. TGF-β signaling, mediated primarily through Smad proteins, regulates cell proliferation, differentiation, and apoptosis [52,53], and its disruption can affect the tumor microenvironment by modulating fibrogenic and proinflammatory cytokines [54].

Beyond direct DNA damage and signaling disruption, tobacco smoke promotes chronic inflammation, a known driver of CRC [41]. Irritants and reactive oxygen species in smoke can activate pro-inflammatory pathways, creating a tumor-promoting microenvironment. Oxidative stress [44] and potential epigenetic alterations further contribute to the carcinogenic effects. In summary, the carcinogenicity of tobacco smoke in the context of CRC arises from the combined action of its numerous components. These agents primarily act by inducing DNA damage and mutations via adduct formation and by dysregulating critical signaling pathways controlling cell proliferation, survival, and the inflammatory response, such as the PI3K/Akt and TGF-β pathways. Understanding these integrated mechanisms underscores the importance of smoking cessation in CRC prevention.

### 3.2. Xenobiotics: Diet, Mycotoxins, and Additives

Studies show that diverse mycotoxins can increase the proliferation and tumor properties of colon cancer cells. In fact, sub-chronic exposure to Ochratoxin A (OTA) significantly facilitates DSS-induced colitis and colitis-associated CRC development [55]. In addition, the mycotoxin zearalenone increases cell proliferation, anchorage-independent growth, and migration of the colon carcinoma cell line HCT116 [56]. Moreover, increased colon cancer cell growth by this mycotoxin occurred through the G protein-coupled estrogenic receptor by promoting G1-to-S phase transition [57]. Interestingly, a low dose of the deoxynivalenol (DON) mycotoxin increased enteritis and CRC in mice [58], although this mycotoxin strongly synergized with AFB1, fumonisin B1(FB1) and OTA for toxicity in Caco-2 intestinal cells [59]. The mechanisms involved included increased intestinal inflammation, altered epithelial cell proliferation (potentially via upregulation of ERK signaling pathway genes promoting proliferation and inhibiting apoptosis), and significant changes in the composition and metabolic activity (specifically fucose and rhamnose metabolism) of the intestinal microbiota.

Some food additives currently used by the general population such as Allura red AC have been suspected of being involved in colon disease and cancer [60,61]. In addition, epidemiological studies suggest that parameters related to sugar consumption are associated with a moderately increased risk of CRC [62]. The presence of trans fats and high levels of saturated fats in processed foods can contribute to chronic inflammation, a recognized risk factor for CRC. Chronic inflammation can lead to the production of pro-inflammatory cytokines and reactive oxygen species (ROS), which can cause DNA damage and promote tumorigenesis [63].

The consumption of red meat (muscle meat from mammals like beef, pork, and lamb) and processed meat (meat transformed through salting, curing, fermentation, smoking, etc.) are among the most well-established dietary risk factors for CRC. Based on extensive evidence, the International Agency for Research on Cancer (IARC) classifies processed meat as “carcinogenic to humans” (Group 1) and red meat as “probably carcinogenic to humans” (Group 2A). The WCRF/AICR concludes there is “convincing” evidence that processed meat causes CRC and “probable” evidence that red meat causes CRC [64]. The carcinogenicity associated with red and processed meat consumption likely arises not from a single factor but from the interplay and potentially synergistic effects of these multiple components and reaction products—the intrinsic heme iron, the additives used in processing (nitrites/nitrates), and the compounds generated during cooking (HCAs/PAHs) [65]. This multi-faceted nature explains the robust association observed epidemiologically. Notably, the mechanism involving heme iron-catalyzed endogenous NOC formation provides a biochemical link explaining why both unprocessed red meat and processed meat (containing added nitrites) increase CRC risk, as both pathways lead to increased exposure to the same class of carcinogenic compounds (NOCs) within the colon [66,67].

### 3.3. Xenobiotics: Pesticides

Pesticides are widely used in agriculture to control pests and enhance crop yields, although the potential link between pesticide exposure and CRC is a growing area of concern. Among them, organophosphorus pesticides (OPs) are known to inhibit the activity of acetylcholinesterase (AChE) by phosphorylating the hydroxyl group of the serine present in the active site of AChE, interrupting the physiological action of AChE, which degrades the neurotransmitter acetylcholine (ACh), causing its accumulation in nerve synapses [68]. This leads to the overstimulation of the muscarinic (mAChR) and nicotinic (nAChR) receptors, resulting in uncontrolled nerve impulses and, consequently, the death of insects [68,69,70,71,72]. However, the toxic effects of OPs do not only affect pests; in fact, all organisms that possess cholinergic components can potentially be affected. While acute toxicity is well-known, chronic exposure to these substances is associated with long-term effects including mutations, epigenetic modifications, and several types of tumors, as well as functional alterations in several physiological systems such as the renal, circulatory, respiratory, endocrine, and immune systems [73,74,75,76,77]. Currently, several non-neuronal cells, such as human breast, pancreatic alpha, endothelial, and placental cell lines express cholinergic components, which make those cells a target for OPs [78,79,80,81,82]. On the other hand, acute OP intoxication has been reported to stimulate the inflammatory response, whereas chronic exposure to low concentrations of OPs increases inflammatory mediators in a sustained manner [83]. Thus, reports indicate that the alteration of the cholinergic system induced by OPs can trigger pathological alterations [83,84,85,86] and has been related to the development of inflammatory diseases such as organophosphate-induced delayed neuropathy [87]. It has been found that the scaffolding protein β-arrestin-1 binds to and activates Scr upon nAChR stimulation via nicotine in non-small-cell lung cancer (NSCLC) and colon cancer cells [87,88]. Acetylcholine (ACh) plays a vital role in various functions in the CNS, but recent studies indicate the importance of ACh in cancer. Apart from being a neurotransmitter, ACh seems to regulate cell proliferation, thus enhancing tumor growth via autocrine and paracrine signaling [89]. Muscarinic subtypes have been shown to have an impact on the development of gastrointestinal cancer [90]. Acetylcholinesterase activity is reduced in tumors, and this reduction is positively correlated with tumor aggressiveness. However, two potent compounds acting as AChE inhibitors, eserine hemisulfate and bis9-amino-1,2,3,4 tetrahydroacridine, resulted in the rapid multiplication of colon cancer cells [91]. Lastly, the development, invasion, and metastasis of colon cancer are all facilitated by M3R activation and post-receptor signaling. Under the EGFR/ERK and PKC/p38 MAPK pathways, M3R signal transduction induces and releases certain matrix metalloproteinases (MMP1, MMP7, and MMP10), which break down the extracellular matrix to promote cell invasion [92]. Oncogenic miRNAs, which already have high basal expression levels, are further elevated in colon cancer cells upon M3R activation. Thus, M3R activation and overexpression appear responsible for the dysregulation of miRNA expression [93].

Organochlorine pesticides (OCPs), a class of persistent environmental pollutants, have been particularly scrutinized due to their potential carcinogenicity. Their long lifespan in the environment and ability to bioaccumulate in the food chain raise concerns about chronic low-dose exposure for humans [94]. The specific mechanisms by which OCPs might promote cancer are still being elucidated, but some known effects offer clues. For instance, DDT and methoxychlor (common OCPs) are neurotoxic, interfering with nerve function by disrupting sodium ion channels in neurons. This disrupts electrical signaling, leading to uncontrolled activity and ultimately cell death [95,96]. This neurotoxic mechanism is the primary reason for the ban on most OCPs in developed countries like the United States [97].

One potential mechanism linking persistent organic pollutants (POPs) to CRC is their lipophilic nature. POPs have an affinity for fat and tend to accumulate in adipose tissue [98]. Over time, this stored fat can release POPs back into the bloodstream, potentially increasing their presence in the colon during excretion. This continuous exposure to POPs in the colon could promote the development of precancerous polyps, which might eventually progress into CRC [99]. Intriguingly, research suggests a two-way interaction between fecal POPs and the gut microbiota. The gut microbiome plays a significant role in CRC development through various mechanisms [100]. Specific bacterial communities within the gut can influence inflammation, immune response, and even the production of carcinogens. Studies suggest that POPs might alter the composition and function of the gut microbiome, potentially creating an environment more conducive to CRC [101]. This complex interplay between POPs and the gut microbiome warrants further investigation to fully understand its role in CRC risk. The potential link between pesticide exposure and CRC requires further research. While the specific mechanisms are not fully understood, the long-lasting presence of POPs in the environment, their potential to accumulate in the colon, and their interaction with the gut microbiome all raise concerns. While environmental exposures like pesticides are implicated, the role of infectious agents, particularly viruses, is another critical aspect of cancer etiology [102]. Table 1 summarizes xenobiotics involved in CRC development.

## 4. Viruses and Cancer

Viral infections are etiologically associated with approximately 13% of the global cancer burden. This percentage is higher in developing countries such as those in Africa and Latin America [103]. Strong evidence demonstrates the role of high-risk human papillomaviruses (HR-HPVs) in cervical, anogenital, and head-and-neck cancers [104]; Epstein–Barr virus (EBV) in lymphomas, nasopharyngeal, and some gastric carcinomas [105]; Hepatitis B and C viruses (HBV, HCV) in hepatocarcinoma; Kaposi Sarcoma virus (KSV) in sarcomas; Merkel cell polyomavirus (MCV) in Merkel cell carcinomas; and Human T lymphotropic virus-1 (HTLV1) in adult T-Cell leukemia. Current research is exploring the potential involvement of these viruses in other human malignancies [106,107]. The potential role of emerging viruses and the human virome in cancer development has been a significant area of research in recent years. For instance, some members of the Polyomaviridae family, such as John Cunningham virus (JCV), are potentially involved in various carcinomas [108]; Murine mammary tumor virus (MMTV) has been associated with certain subtypes of breast cancer [109]; and Endogenous retroviruses (ER) have been associated with additional types of cancer [110].

Although viruses have been identified in cancer patients, healthy individuals harbor a diversity of viruses, the human virome, which shows a remarkable heterogeneity. The human virome is extremely broad and complex, composed of approximately 10^13^ particles per individual, which can influence the regulation of immune homeostasis through interactions with bacteria and the host immune system [111]. Several viruses have been identified in the normal organism, given that different tissues and organs house diverse viral communities. In addition, the virome is an integral part of the microbiome, and when evaluating its impact on human health, all its components must be considered as an entity, together with the interactions established with the host [112]. Approaches based on RNA/DNA next-generation sequencing (NGS) have shown that within the human body, there is a large diversity of viruses, some without known clinical or biological significance [113]. In health conditions, the human intestine generally presents low levels of eukaryotic viruses, although viral DNA lineages can be detected, including members of the Anelloviridae, Geminiviridae, Herpesviridae, Nanoviridae, Papillomaviridae, Parvoviridae, Polyomaviridae, Adenoviridae, and Circoviridae. Among the ubiquitous members of the human virome, anelloviruses present a unique case due to their high prevalence and unclear pathogenicity, making their potential role in CRC particularly intriguing [114].

## 5. Anelloviruses: Biology and Potential Role in Colorectal Cancer

Anelloviruses (AVs), members of the Anelloviridae family (so-called the anellome in humans), are an emerging family of ubiquitous viruses found in the general population [15]. AVs are very small, naked viruses, with a size of approximately 45 nm in diameter, a negative-sense single-stranded circular DNA genome ranging from 2.0 to 3.9 Kb [115], and a capsid with icosahedral symmetry and 12 capsomers. The DNA genome is organized into two regions, a coding region including three to five open reading frames (ORFs1–5) and a non-coding region rich in G/C content (UTR), with regulatory functions [116]. The highly variable Anelloviridae family currently encompasses 30 genera and 155 species, which have possibly evolved through intense recombination events [115,117]. Among these genera, three are known to infect humans: Alphatorquevirus, Betatorquevirus, and Gammatorquevirus, which are conventionally called Torque Teno Virus (TTV), Torque Teno Mini Virus (TTMV), and Torque Teno Midi Virus (TTMDV), respectively. Alphatorquevirus encompasses 29 species (TTV 1 to 29), Betatorquevirus covers 12 species (TTMV 1 to 12), and Gammatorquevirus includes 2 species (TTMDV 1 and 2) [118,119] (Figure 1). It has been suggested that TTV is the predominant virus of this family in the adult population, even though the composition of the anellome varies during an individual’s lifetime [120]. Torque Teno Virus (TTV) was initially detected in 1997 in a Japanese patient with acute post-transfusion hepatitis of unidentified cause [121]. Indeed, certain TTV antigens have been detected in systemic lupus erythematosus (SLE) and multiple sclerosis (ORF2 and N-arginine terminal regions), suggesting a potential involvement in autoimmune disorders [122]. Furthermore, the viral load has been found to be higher in patients with idiopathic pulmonary fibrosis (IPF) and lung cancer [123]. Similarly, TTV has been detected in higher percentages in children with bronchopneumonia than in children with mild acute respiratory diseases [124]. It has been suggested that this virus could contribute to lung damage caused by asthma in children [125]. Additionally, TTV infection has been reported in 77.5% of patients with prolonged sepsis [126]. Similarly, a 71.5% prevalence of TTV has been reported in individuals with some alteration in carbohydrate metabolism or some type of inflammatory disease, with positivity being observed to be higher in patients with hypertension, type 2 diabetes mellitus, and patients with breast cancer [127]. Of note, a high prevalence of AVs has been observed in immunocompromised individuals, such as lung transplant recipients, patients infected with human immunodeficiency virus (HIV), and individuals receiving treatment with immunosuppressive compounds due to intestinal inflammatory diseases. This finding suggests that the Anelloviridae family is regulated by the host’s immune response [128,129,130,131]. A role for the anellome in human diseases has not been established, although diverse studies have associated the anellome with specific human alterations, including cancer [132]. Indeed, AVs are regarded as “viruses awaiting a disease”.

Even though AVs are pantropic, the study of their replication has been hampered by the absence of an efficient cell system for viral replication in vitro. Thus, the mechanism of AV replication is not fully understood. However, evidence from other single-stranded DNA viruses suggests that AVs replicate within the host cell nucleus, utilizing the host enzymatic machinery, in a rolling circle replication process [133]. The untranslated regions (UTRs) flanking the viral genes contain specific sequences that fold into hairpin structures. These hairpins play a crucial role in viral replication by facilitating a rolling-circle mechanism, by which the viral genome is continuously copied in a circular fashion [134]. The N-terminus of the AVs ORF1 protein contains an arginine-rich region resembling the viral genome-binding ARM motif. This structural homology suggests that the ORF1 protein might be involved in AV genome packaging and replication processes [120]. Recently, after an in-depth analysis of the ORF1 region of AVs and their different genera, a remarkable variation in size was observed due to insertions in the jelly-roll (JR) domain of the virus capsid protein [134]. In fact, JR had previously been described as a 60-mer icosahedral particle, extending to the spike domains, forming a crown-like protruding surface, with more conserved amino acids found at the base of the spike and the most hypervariable region at the apex (P2 region covering the conserved P1 spike domain). This suggests that this same conformation would allow evasion of the host immune response (very diverse epitopes at the apex and conserved ones protected at the base) [135].

AVs exhibit a unique replication strategy. Their single-stranded DNA genome is transcribed into only three mRNAs. However, through a process likely involving alternative splicing or ribosomal frameshifting, these three mRNAs are translated into at least six distinct proteins [136]. The ORF3 protein possesses a serine-rich region and a C-terminal cluster of arginine and lysine residues, indicative of a potential nuclear localization signal. Additionally, the presence of multiple nuclear targeting sequences suggests its role in transcriptional regulation [137]. Furthermore, its homology to DNA topoisomerase I further supports a potential role for ORF3 in viral replication and host interactions [138].

Evidence suggests Anelloviruses (AVs) transmit person-to-person through diverse routes. For instance, AVs have been detected in 93% of saliva samples and 38.8% of plasma from patients with liver cirrhosis [139], while breast milk represents another potential route, with AV DNA identified in 23.3% of samples [140]. Torque Teno Virus (TTV), a specific AV, demonstrates widespread presence across various bodily fluids and tissues. It has been reported in 60% of semen samples from drug addicts [141], the nasal mucosa of children with acute respiratory disease (ARD) [124], and a range of extracellular fluids in Japan, including tears, sweat, urine, feces, saliva, and serum [142]. High prevalence is also noted in bile, with TTV DNA found in 84.6% of individuals undergoing biliary fluid drainage [143]. Furthermore, TTV is harbored within the hematopoietic and lymphatic systems, detected in bone marrow cells (BMCs), peripheral blood mononuclear cells (PBMCs) [144] as messenger RNA (mRNA) within bone marrow [145], and in lymph nodes of patients with Hodgkin’s lymphoma (52%) and nonspecific lymphoid hyperplasia (60%) [146]. The liver is another significant site, with TTV detection rates varying by condition: 73.6% in acute viral hepatitis (AVH), 59.2% in fulminant liver failure (FHF), 21.5% in chronic active hepatitis (CAH), and 29.1% in liver cirrhosis (LC) [147]. Beyond these locations, evidence from China indicates vertical transmission is possible, with TTV prevalence reaching 17.8% in pregnant women and a transplacental transmission rate of 13.8% [148]. Notably, viral titers in saliva are often significantly higher, observed to be 100 to 1000 times greater than those in plasma [149].

A pathogenic role for the anellome in CRC has not been established. However, associations have been found through epidemiological approaches. In 2002, TTV was detected for the first time as being more frequent in the nucleus of PBMCs from patients with cancer, including CRC, when compared to control patients [150]. In the same year, de Villiers et al. analyzed human cancers including colon carcinomas and colon polyps. TTV-related sequences were found in 9 out of 13 (69.2%) colon carcinomas and in 4 out of 7 (57.1%) colon polyps. In addition, a high heterogeneity of TTV sequences was observed in tumors [151]. Later, in 2007, De Villiers et al. suggested a preferential presence of TTV in CRC when compared to adjacent non-tumor tissues, although causality was not demonstrated [152]. In 2023, using metagenomic approaches, it was found that mucosal virome dysbiosis in CRC patients from China was dominated by non-bacteriophages. Moreover, data demonstrated that the virome was dominated by eukaryotic viruses, especially human anelloviruses [153]. Interestingly, a study suggested the importance of the eukaryotic gut virome, including the Anelloviridae family, during the pathogenesis of Crohn’s disease, a type of inflammatory bowel disease [154]. Taken together, the possibility of AVs being etiologically involved in CRC is plausible, though additional studies are warranted (Figure 2).

## 6. Perspectives

Anelloviruses (AVs) are a ubiquitous component of the human virome. They are commonly found in both health and disease. This ubiquity suggests that their potential contribution to pathology, including colorectal cancer (CRC), likely requires specific host or environmental cofactors. Immunosuppression is one recognized host factor that can enhance the pathogenicity of opportunistic agents [155]. On the other hand, xenobiotics represent a diverse class of environmental factors known to influence host physiology, including immune function and carcinogenesis [156,157].

Research is ongoing into the individual roles of AVs and certain xenobiotics in CRC. However, the potential for synergistic interaction between these factors remains largely unexplored and speculative. This interaction might occur particularly through modulation of the immune system. Therefore, it is crucial to critically synthesize the current knowledge and propose specific hypotheses regarding immune-mediated crosstalk between xenobiotics and AVs in the context of CRC. It is also important to outline targeted future research directions focused on this interaction. To understand the potential for interaction, the focus must be on how both xenobiotics and AVs interface with the immune system, particularly within the colonic microenvironment relevant to CRC. Numerous xenobiotics, such as components of tobacco smoke, certain pesticides, and various dietary factors, possess immunomodulatory properties. These effects can range from directly suppressing immune cell function and altering gut barrier integrity to modifying the gut microbiota and inducing DNA damage. Specifically, alterations in key immune pathways involved in inflammation and pathogen recognition are critical considerations when evaluating xenobiotic impact [158]. Concurrently, AVs interact dynamically with the host immune system. For example, their prevalence increases in immunosuppressed individuals. AVs may possess immunomodulatory capabilities themselves, potentially contributing to chronic low-grade inflammation or altering local immune cell phenotypes [159]. The reactivation of AVs, perhaps facilitated by xenobiotic-induced immunosuppression, could further amplify these effects. This interplay occurs against the backdrop of the CRC immune landscape, which is characterized by chronic inflammation, altered immune cell infiltrates, and mechanisms of immune evasion—a setting potentially exacerbated by combined xenobiotic and AV influences. Currently, direct evidence for cooperation between specific xenobiotics and AVs in CRC pathogenesis via immune mechanisms is lacking. Furthermore, the ubiquitous nature of AVs complicates establishing causality. While existing knowledge on DNA virus-xenobiotic co-carcinogenesis models offers a framework, significant gaps remain, particularly concerning how specific xenobiotic exposure affects AV replication dynamics in vivo and how their combined presence impacts specific colonic immune cell populations and effector functions. Based on the intersection of known effects, specific, testable hypotheses centered on immune pathways can be formulated. For instance, one hypothesis is that certain xenobiotics suppress local antiviral immune responses (e.g., Type I Interferon signaling) in the colonic mucosa. This suppression could facilitate increased AV replication, leading to chronic immune stimulation via alternative pathways (e.g., NF-κB activation). This, in turn, could promote a pro-tumorigenic inflammatory microenvironment. Another hypothesis posits that xenobiotic-induced gut dysbiosis creates an altered immune milieu. This might enhance AV persistence or modulate the host response to AVs, potentially skewing T-cell responses and contributing to immune evasion by nascent CRC cells.

Moving beyond speculation requires a focused research strategy incorporating specific methodological approaches that target the proposed immune interactions. Advanced in vitro studies should employ co-culture systems using human colonic organoids, relevant immune cells (macrophages, T-cells, and dendritic cells), and potentially commensal bacteria. These systems could be exposed to specific, relevant xenobiotics +/−AV infection (using available replication models or viral components) [160]. Such studies would help dissect mechanisms by analyzing viral replication levels (qPCR), cell viability, barrier function (TEER measurements), detailed cytokine/chemokine profiles (multiplex assays and ELISA), immune cell activation markers (flow cytometry), and the activation status of key signaling pathways (Western blot, phosphoprotein assays, and qPCR for pathway targets). Additionally, in vivo studies are essential. These could utilize appropriate mouse models, potentially gnotobiotic mice colonized with specific AV strains and relevant gut microbiota, exposed to controlled doses of key xenobiotics. These studies should monitor AV loads and perform detailed immune cell profiling (using flow cytometry, or single-cell RNA-seq) in gut-associated lymphoid tissues and tumors. They should also measure local and systemic cytokine levels, assess gut barrier integrity, and evaluate CRC development/progression in established cancer induction models. Furthermore, integrated human cohort studies are crucial. These studies should particularly focus on populations at risk, including those with rising early-onset CRC incidence. These studies must collect detailed data on xenobiotic exposure (using validated questionnaires and specific exposure biomarkers in blood/urine/stool), dietary habits, and gut microbiome composition (16S rRNA sequencing, metagenomics). Data should also include AV presence/load/genotype (using sensitive molecular methodologies) and comprehensive immune profiling (circulating immune markers, cytokine panels, and where feasible, tissue-level immune analysis from biopsies/resections).

Focusing future research on the specific immune pathways mediating this crosstalk offers a promising avenue. This focus should employ the detailed methodological approaches outlined above, offering a promising avenue. Elucidating these mechanisms, driven by clear hypotheses and utilizing advanced experimental models and integrated human studies, is essential to move from speculation to understanding the potential cooperation between xenobiotics and AVs in CRC (Figure 3). Such efforts could yield crucial insights into CRC etiology and potentially reveal novel targets for prevention or therapy. Ultimately, this research could address the intricate interplay of environmental factors, the virome, and host immunity in cancer.

## 7. Conclusions

The development of colorectal cancer is increasingly understood as a multifactorial process influenced by genetic predisposition, lifestyle, and environmental exposures. Among environmental factors, xenobiotics, particularly those found in tobacco smoke, play a well-documented role. These compounds act as potent carcinogens, primarily by inducing DNA damage through adduct formation and subsequent mutations, disrupting critical cellular signaling pathways like PI3K/Akt and TGF-β, promoting oxidative stress, and fostering chronic inflammation. This cascade of molecular events contributes significantly to genomic instability and creates a microenvironment conducive to neoplastic transformation in the colon.

Concurrently, the human body is persistently colonized by commensal viruses, such as AVs. As highlighted, AVs exhibit remarkable ubiquity, detected in a vast array of bodily fluids and tissues, from saliva and blood plasma to bile, lymphoid tissue, and the liver itself. While often considered non-pathogenic bystanders, their high prevalence, particularly noted in states of inflammation or existing disease (e.g., liver conditions or lymphoid hyperplasia), raises questions about their potential role as modulators of host physiology or pathology.

Considering these distinct factors, a potential interaction in CRC development warrants investigation. While xenobiotics like those in tobacco smoke provide direct carcinogenic insults, the ubiquitous presence of AVs introduces another layer of complexity. It is conceivable that the chronic inflammation, tissue damage, and immune alterations induced by xenobiotic exposure could create a more permissive environment for AV replication or persistence within the colorectal mucosa or associated tissues. Conversely, a high AV load or specific AV genotypes might modulate the host’s immune response to xenobiotic-induced damage or pre-cancerous lesions, potentially exacerbating pro-tumorigenic inflammation or impairing effective immune surveillance. Although direct evidence for a synergistic interaction between specific xenobiotics and AVs in CRC pathogenesis is lacking, their coexistence and independent associations with cellular stress and inflammation suggest that exploring their combined impact could reveal novel insights into CRC etiology and risk stratification. Further research is essential to elucidate whether AVs act merely as markers of exposure or underlying conditions, or if they actively participate, alongside xenobiotic pressures, in the complex journey towards colorectal malignancy.

## Figures and Tables

**Figure 1 ijms-26-04354-f001:**
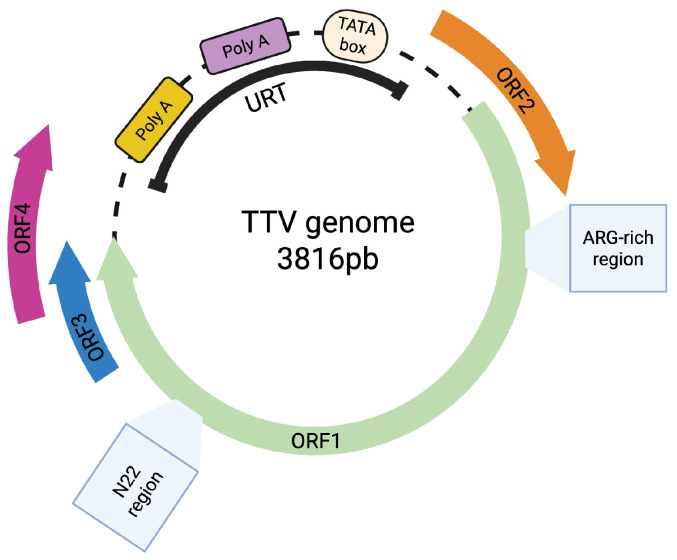
Torque Teno Virus genome organization. ORF: Open reading frame.

**Figure 2 ijms-26-04354-f002:**
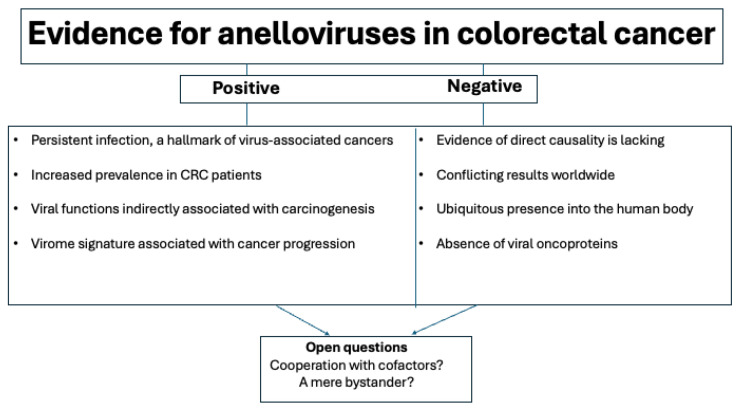
The evidence supporting or not a role for anelloviruses in colorectal carcinogenesis.

**Figure 3 ijms-26-04354-f003:**
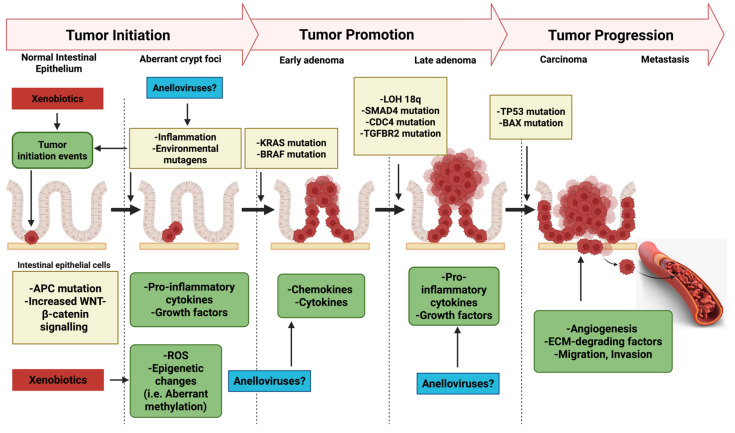
A CRC model where the possible cooperation between xenobiotics and anelloviruses is proposed, suggesting mechanisms involved during tumor initiation and promotion. Prepared by BioRender (https://www.BioRender.com/).

**Table 1 ijms-26-04354-t001:** Xenobiotics involved in colorectal cancer.

Xenobiotic Source/Class	Examples	Potential Mechanisms in CRC	Ref
Tobacco smoke		Formation of DNA adducts, induction of mutations, promotion of inflammatory processes. Deregulation of multiple signaling pathways (e.g., PI3K/Akt, TGF-β).	[41,48]
	NNK	Potent nitrosamine: Metabolic activation; DNA adducts; mutations.	[42,43,45]
	Benzopyrene (BaP)	Forms stable DNA adducts; mutations in oncogenes/tumor suppressor genes; Induces oxidative stress (OS); DNA damage.	[44,48,49]
	Aromatic amines (AaC)	Forms DNA adducts; Induces aberrant crypt foci (animal models).	[46,47]
	Nitrosamines	Metabolic activation; reactive species; DNA damage, DNA adduct formation, disruption of cellular signaling pathways.	[42,43,44]
Mycotoxins	Ochratoxin A (OTA)	Facilitates DSS-induced colitis and colitis-associated CRC development.	[55]
	Zearalenone	Increases proliferation, anchorage-independent growth, migration (HCT116 cells); Acts via G protein-coupled estrogenic receptor; Promotes G1-to-S phase transition.	[56,57]
	Deoxynivalenol (DON)	Increased enteritis and CRC (mice, low dose); Synergistic toxicity with other mycotoxins (Caco-2 cells).	[58,59]
Food additives	Allura red AC	Suspected involvement in colon disease and cancer.	[60,61]
Processed foods	Trans fats, saturated fats	Contribute to chronic inflammation (recognized CRC risk factor); pro-inflammatory cytokines, ROS; DNA damage, tumorigenesis.	[63]
Processed meats	(contains preservatives, fats)	Consistently linked to increased CRC risk.	[64,65,66,67]
Sugar		Epidemiological association between sugar consumption parameters and moderately increased CRC risk.	[62]
Pesticides		General: Potential link is a growing concern.	
Organophosphorus (OPs)		Inhibit Acetylcholinesterase (AChE); ACh accumulation; overstimulation of cholinergic receptors; Chronic exposure: mutations, epigenetic modifications, tumors, inflammation, alteration of cholinergic system linked to pathological changes & inflammatory disease.	[68,69,70,71,73,74,75,76,77]
		Potential indirect effects via ACh regulation of cell proliferation, β-arrestin-1/Src activation, M3R signaling impacting MMPs and miRNAs.	[87,88,89,91,92,93]
Organochlorines (OCPs)	DDT; methoxychlor	Persistent, bioaccumulate; Potential carcinogenicity; Neurotoxic mechanism (sodium channel disruption) known but link to cancer unclear.	[94,95,96,97]
Persistent Organic Pollutants (POPs—includes some pesticides)		Lipophilic; accumulate in adipose tissue; potential release & colon exposure; May alter gut microbiota composition/function; pro-CRC environment.	[98,99,101,102]

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
