# Peer review of "The Role of Xenobiotics and Anelloviruses in Colorectal Cancer: Mechanisms and Perspectives"

_ijms, 2025, doi:10.3390/ijms26094354_

Round 1

Reviewer 1 Report

Comments and Suggestions for Authors

The authors summarize the role of Xenobiotics and Anelloviruses in CRC. Some concerns should be addressed before potential publication.

  1. Please supplement a table that conclude the respective mechanism and Xenobiotics such as smoke components, diet mycotoxins, pesticides.
  2. Information of Figure 1 is too simple, they can strengthen the classification and evolutionary trajectories with the nearest virus that function have been clearly elaborated.
  3. I advise the author add the distribution of anellovirus in heathy people, adenoma, CRC to determine whether an enrichment of anellovirus in CRC or not. In addition, is anellovirus infection associated with a poor outcome or metastasis? Then, does target anellovirus a novel treatment mean?
  4. The resolution of Figure 3 should be improved.

Author Response

Reviewer 1

The authors summarize the role of Xenobiotics and Anelloviruses in CRC. Some concerns should be addressed before potential publication. Please supplement a table that conclude the respective mechanism and Xenobiotics such as smoke components, diet mycotoxins, pesticides.

Answer

Many thanks for this suggestion. A table was created, using the information describe in this manuscript.

Reviewer

Information of Figure 1 is too simple; they can strengthen the classification and evolutionary trajectories with the nearest virus that function have been clearly elaborated.

Answer

Many thanks for this suggestion. The objective of this figure is to show the genome organozation of anelloviruses, which is important for its functionality. So, the previous figire was replaced by another which clarify the genome structure and organization.

Reviewer

I advise the author to add the distribution of anellovirus in heathy people, adenoma, CRC to determine whether an enrichment of anellovirus in CRC or not.

Answer

Many thanks for this observation. Studies considering the prevalence of anellovirus in control or individuals without cancer were included in this revision. However, additional descriptions regarding such studies, were included (Ref. 150-153).

Reviewer

In addition, is anellovirus infection associated with a poor outcome or metastasis? Then, does target anellovirus a novel treatment means?

Answer

Many thanks for this question. Information regarding this point is lacking, so we can not speculate respect a potential association with clinical outcome or importance in CRC treatment.

Reviewer

The resolution of Figure 3 should be improved.

Answer

This was improved.

Reviewer 2 Report

Comments and Suggestions for Authors

Chapter 3.1 In this section on tobacco smoke components and CRC, the structure is repetitive. Each compound is introduced with the same sequence: DNA adduct formation, some functional alteration, and CRC relevance. While the information is relevant, this pattern makes the section feel redundant. I recommend a more integrated approach—summarizing shared mechanisms across compounds, including pathways affected, and only highlight specific differences or particularly strong evidence where applicable.

 Next, the authors refer back to signaling pathways introduced in the previous section but then expand on them by introducing additional pathways here. This makes the logic of the chapter unclear. This includes also an unexpected mention of the effects of aspirin and lycopene as tumour supressors... (lines 139-143)

The very same repetitivness can be found in chapter 5, when introducing transmission routes of AVs, with the same type of phrase repeated on where the viruses were found: "In addition, Likevise, ...also been found, Similarly, Similarly, Finally, In addition to the above....."

The individual chapters of the manuscript read often more like encyclopaedic list of isolated facts than a critical review that provides added value. Most of the information could be more effectively presented in summary tables, while the text should focus more on the extraction of patterns, knowledge gaps, or potential hypotheses.

From the perspective of the whole manuscript, it fails to bring the promised integration of xenobiotics and anelloviruses in CRC.  The chapters on CRC development, Xenobiotics in CRC and Anelloviruses in CRC read more like three small mini-reviews. There is a disbalance in the subchapters of the Xenobiotics chapter. Some, like the one dedicated to tobacco smoke, are relatively extensive, while dietary and mycotoxin chapter is very short and far from comprehensive.

The final chapter Perspectives  does not bring these chapters together in a meaningful way. The possibility of interaction between xenobiotics and anelloviruses in CRC are briefly mentioned, but all this remains vague and speculative.

I would recommend to perform a more thorough critical assessment of the existing knowledge with more focus on potential future directions. The content of the previous chapters could be focused/restricted only on the critical know how that is important. For instance from the perspective of the interaction of the xenobiotics and anelloviruses with immune system and the involved pathways. 

Minor issues - the manuscript comprises a number of typos and would benefit from a more thorough proofreading. To select a few:

    • line 15: diases
    • line 43: diet factors (dietary?)
    • Line 57: Patways
    • line 71: demonsttated and trancritional
    • line 83: downr egulation
    • line 118: thys, ...

Author Response

Chapter 3.1 In this section on tobacco smoke components and CRC, the structure is repetitive. Each compound is introduced with the same sequence: DNA adduct formation, some functional alteration, and CRC relevance. While the information is relevant, this pattern makes the section feel redundant. I recommend a more integrated approach—summarizing shared mechanisms across compounds, including pathways affected, and only highlight specific differences or particularly strong evidence where applicable. Next, the authors refer back to signaling pathways introduced in the previous section but then expand on them by introducing additional pathways here. This makes the logic of the chapter unclear. This includes also an unexpected mention of the effects of aspirin and lycopene as tumour supressors... (lines 139-143)

Answer

Many thanks for these observations; the manuscript was completely modified and corrected. This section was improved, according to these comments.

Reviewer

The very same repetitivness can be found in chapter 5, when introducing transmission routes of AVs, with the same type of phrase repeated on where the viruses were found: "In addition, Likevise, ...also been found, Similarly, Similarly, Finally, In addition to the above....."

Answer

This was corrected.

Reviewer

The individual chapters of the manuscript read often more like encyclopaedic list of isolated facts than a critical review that provides added value. Most of the information could be more effectively presented in summary tables, while the text should focus more on the extraction of patterns, knowledge gaps, or potential hypotheses.

Answer

Many thanks for this observation. A table including xenobiotics and role in CRC was included.

Reviewer

From the perspective of the whole manuscript, it fails to bring the promised integration of xenobiotics and anelloviruses in CRC. The chapters on CRC development, Xenobiotics in CRC and Anelloviruses in CRC read more like three small mini reviews. There is a disbalance in the subchapters of the Xenobiotics chapter. Some, like the one dedicated to tobacco smoke, are relatively extensive, while dietary and mycotoxin chapter is very short and far from comprehensive.

Answer

Many thanks for this important observation. In this reviewed manuscript, we described separately both factors (AVs and xenobiotics) and in the perspective, we suggest future research in this field. This is because an interaction between both factors has never been addressed. Additionally, the chapters were balanced.

Reviewer

The final chapter Perspectives does not bring these chapters together in a meaningful way. The possibility of interaction between xenobiotics and anelloviruses in CRC are briefly mentioned, but all this remains vague and speculative. I would recommend performing a more thorough critical assessment of the existing knowledge with more focus on potential future directions. The content of the previous chapters could be focused/restricted only on the critical know how that is important. For instance, from the perspective of the interaction of the xenobiotics and anelloviruses with immune system and the involved pathways. 

Answer

The manuscript was completely checked and improved. These concerns were considered and improved. A potential cooperation or synergism between AVs and xenobiotics was included as a perspective, which was improved. At this moment, this potential interaction remains speculative, although we suggest this as an interesting focus for future research.

Reviewer

Minor issues - the manuscript comprises a number of typos and would benefit from a more thorough proofreading. To select a few:

    • line 15: diases
    • line 43: diet factors (dietary?)
    • Line 57: Patways
    • line 71: demonsttated and trancritional
    • line 83: downr egulation
    • line 118: thys, ...

Answer:

These issues were corrected. Additional typographic mistakes were corrected in the manuscript.

Round 2

Reviewer 2 Report

Comments and Suggestions for Authors

The authors addressed most of my concerns and improved the manuscript, making the text flow better, revising the stylistics, including a summary table. Most importantly, the Perspectives section has been substantially improved.

A few concerns, however, remain:

- In the first review, I noted that the manuscript introduced additional CRC-related pathways later in the text, disrupting the logical flow. Although the authors stated this was corrected, the issue persists. Several important pathways (e.g., TGF-β, ERK, MAPK, cholinergic signaling) are still introduced later without having been mentioned in the initial dedicated chapter on CRC pathways. Given that the manuscript already includes a full chapter on CRC signaling pathways, introducing new ones later without prior mention is not justified. I recommend expanding the initial overview to include all pathways discussed later, or, at minimum, adding a statement at the end of the chapter indicating that additional pathways will be introduced as contextually relevant in subsequent sections.

- In the Perspectives section, several sentences are overly long and dense, making the text difficult to follow in places. I recommend editing this section for clarity by shortening sentences and splitting complex ideas into more manageable units. An example phrase is:

"Numerous xenobiotics, including components of tobacco smoke, certain pesticides, and dietary factors, possess immunomodulatory properties, ranging from direct suppression of immune cell function and alteration of gut barrier integrity to modification of the gut microbiota and induction of DNA damage; specifically, alterations in key immune pathways involved in inflammation and pathogen recognition are critical considerations [158]. "

Minor identified issues:

- Figure 3 is not referenced anywhere in the main text
- Table 1 does not include references for the statements presented

Author Response

A few concerns, however, remain:

  • In the first review, I noted that the manuscript introduced additional CRC-related pathways later in the text, disrupting the logical flow. Although the authors stated this was corrected, the issue persists. Several important pathways (e.g., TGF-β, ERK, MAPK, cholinergic signaling) are still introduced later without having been mentioned in the initial dedicated chapter on CRC pathways. Given that the manuscript already includes a full chapter on CRC signaling pathways, introducing new ones later without prior mention is not justified. I recommend expanding the initial overview to include all pathways discussed later, or, at minimum, adding a statement at the end of the chapter indicating that additional pathways will be introduced as contextually relevant in subsequent sections.
  • Answer: Many thanks for this observation. This was corrected.
  • In the Perspectives section, several sentences are overly long and dense, making the text difficult to follow in places. I recommend editing this section for clarity by shortening sentences and splitting complex ideas into more manageable units. An example phrase is:

    "Numerous xenobiotics, including components of tobacco smoke, certain pesticides, and dietary factors, possess immunomodulatory properties, ranging from direct suppression of immune cell function and alteration of gut barrier integrity to modification of the gut microbiota and induction of DNA damage; specifically, alterations in key immune pathways involved in inflammation and pathogen recognition are critical considerations [158]. "
  •  
  • Answer. This was corrected.

    Minor identified issues:
  • Figure 3 is not referenced anywhere in the main text
    - Table 1 does not include references for the statements presented

Answer: This was corrected. Many thanks for these observation to improve the manuscript.